# Vitamin D Attenuates Ulcerative Colitis by Inhibiting ACSL4-Mediated Ferroptosis

**DOI:** 10.3390/nu15224845

**Published:** 2023-11-20

**Authors:** Shuo Gao, Can Sun, Juan Kong

**Affiliations:** Department of Clinical Nutrition, Shengjing Hospital of China Medical University, Shenyang 110004, China; lovegaoshuo@163.com (S.G.); my1234756@163.com (C.S.)

**Keywords:** vitamin D, ferroptosis, ACSL4, Ulcerative Colitis

## Abstract

Background: With environmental and lifestyle changes, recent epidemiological studies have shown that the prevalence of Ulcerative Colitis (UC) is on the rise, while treatment options are limited. There is an urgent need to explore the underlying mechanisms of vitamin D (VD) as an effective treatment. Methods: Dextran sulfate sodium-induced mice and lipopolysaccharide-induced HCT116 cells were used to establish the classic UC models in vivo and in vitro, respectively. Typical symbols of inflammation (IL-6, COX-2), oxidative stress (MDA, MPO, GSH), and ferroptosis (ACSL4, GPX4, SLC7A11, and Iron) were analyzed by Western blot, Immunohistochemistry, RT-PCR, and relative assay kits. The inflammation factors and oxidative stress injury of cells transfected with ACSL4^+/+^ plasmids were tested by Western blot, MDA, and MPO methods. Results: Vitamin D attenuated the levels of COX-2, IL-6, Iron, MDA, and MPO and improved SOD1 and GSH contents in DSS + VD and LPS + VD groups, compared with model groups. Ferrostatin-1 (Fer-1) could relieve the levels of COX-2, IL-6, Iron, MDA, and MPO while increasing the contents of SOD1 and GSH in DSS + Fer-1 and LPS + Fer-1 compared to model groups. VD downregulated the expression of ACSL4 and upregulated GPX4 in tissues and cells. After transfected with ACSL4^+/+^ plasmids, we found VD’s role of downregulating inflammation and oxidative stress was relieved. Conclusions: Vitamin D can relieve UC by inhibiting ferroptosis both in mice and in cells through the negative regulation of ACSL4, providing new insight into the therapeutic function of VD on UC.

## 1. Introduction

Inflammatory bowel disease (IBD) is a chronic disease [1] that seriously threatens people’s health and severely impacts economic development worldwide. IBD, which mainly includes Ulcerative Colitis (UC) and Crohn’s disease (CD) [2], is characterized by repetitive episodes of inflammation of the gastrointestinal tract (GIT) caused by an abnormal immune response to gut microflora. UC, a nonspecific intestinal inflammatory disease with a prolonged duration, was involved in consecutive and diffuse enteric mucosal damage. Diarrhea, bloody stools, and abdominal pain are the typical clinical features. The pathogenesis of UC involves multiple aspects, including gene polymorphism, immune response, oxidative stress, and microbial infection [3]. A combination of corticosteroids, immunosuppressors, and biological therapies are currently used in clinical treatment to induce remission [4]. However, the specific pathological process of UC has not been well verified [5]. Thus, it is essential to explore effective therapy strategies for UC.

Vitamin D receptor (VDR) is a member of the nuclear hormone superfamily, which exists in almost all tissues of the human body. VD regulation can promote the absorption rate of calcium and phosphorus in intestinal tissues and increase the number of osteoblasts in its classic role. VD is required to bind to VDR to function in the body. However, the immunomodulatory effects of VD to improve autoimmune diseases are still unclear [6]. VD has been proven to inhibit the process of inflammation in several ways: regulating autophagy, reducing oxidative stress [7], reducing leukocyte differentiation and activation [8], and enhancing tight junctions in intestinal epithelial cells. A large amount of epidemiological and laboratory evidence suggests that VD is involved in the pathogenesis process of UC. First, research indicated the serum levels of VD in UC adults are lower than those in healthy people. Second, low VD levels can negatively affect the intestinal barrier and immune system function, thereby affecting the pathogenesis and progression of UC [9,10]. However, the potential mechanism of VD mitigating UC is still unclear and needs to be further studied.

Ferroptosis was defined as a unique form of non-apoptotic cell death mode in 2012, characterized by the oxidation of polyunsaturated fatty acids (PUFAs) containing phospholipids, the presence of redox-active Iron, and loss of lipid peroxide repairing ability. It can help maintain the balance of death in normal cells and tissues and is involved in the pathological process of stroke, cerebral hemorrhage, Huntington’s disease, cancer, renal ischemia-reperfusion injury, and other diseases [11]. The biochemical mechanism underlying ferroptosis is the Iron-catalyzed formation of lipid radicals combined with the depletion of glutathione (GSH) or the inactivation of the lipid repair enzyme GSH peroxidase 4 (GPX4) [12,13]. Much evidence demonstrates that ferroptosis can be inspired by the downregulation of system Xc-activity (XcT), also known as SLC7A11; SLC7A11 (Solute Carrier Family 7 Member 11) is the 11th member of the solute transport 7 family. It belongs to the cystine/glutamate antiporter protein and mainly participates in the transport of amino acids on the plasma membrane [13,14]; it also includes inhibition of GPX4 and the increase in lipid ROS [11,15]. Previous studies suggested that sensitivity to erastin-induced ferroptosis was found in the LPS-induced cell model and could be relieved by a ferroptosis-specific inhibitor: Ferrostatin-1(Fer-1). LPS activates gene expression of inflammatory cytokines and induces inflammatory responses by binding to cell surface receptors. Also, a similar phenomenon has been presented in the DSS-induced mice model [16,17].

Acyl-CoA synthetase long-chain family member 4 (ACSL4), as a member of the long-chain acyl coenzyme A synthase family (ACSLs), plays an essential role in the induction of ferroptosis. Unlike other family members, ACSL4 catalyzes the synthesis of arachidonic acid (AA) into arachidonic acid coenzyme A, which takes part in the synthesis of membrane phospholipids. PUFAs are involved in a variety of processes, including membrane phospholipid composition, the synthesis of lipid signaling pathways and conduction of ferroptosis signaling of lipid oxidation, and other processes to induce intracellular ferroptosis [18,19]. ACSL4 is considered a sensitive monitor and an essential contributor to ferroptosis, although its specific role in UC remains unknown. Hence, there is an urgent need for a better understanding of the functions of ACSL4 in UC and its relationship with ferroptosis.

In a previous study by our team, Xiong [20] confirmed that VD can effectively alleviate DSS-induced UC by downregulating the expression of pro-inflammatory factors while maintaining the integrity of the intestinal mucosal epithelium. However, whether VD can attenuate UC by inhibiting ACSL4-mediated ferroptosis is still unknown. In the current research, we suppose that ACSL4 may be an important target of VD to mitigate UC, and the supplementation of VD can ameliorate the procedure of ferroptosis and defer the development of UC.

## 2. Materials and Methods

### 2.1. Animal Treatment

Male ICR mice (6 weeks old) were purchased from Liaoning Changsheng Biotechnology (Benxi, China) and reared in the Animal Facility of Shengjing Hospital of China Medical University (Shenyang, China) [21]. After acclimatizing for seven days, 30 mice (n = 5 per group) were randomly divided into the following 6 groups, including Control (Con), VD, Fer-1, DSS, DSS + VD, and DSS + Fer-1 groups. We performed a total of three replicate experiments. All the experimental procedures were approved by the Institutional Ethics Committee of China Medical University. To induce UC, animals were given 3% DSS (R051515, 50,000 MW, RHAWN, Shanghai, China) in the drinking water for 10 days for the DSS, DSS + VD, and DSS + Fer-1 groups. VD and DSS + VD groups were given CCE (10 μL in 100 mL of water) [20], a VD analog, for 2 weeks [22]. Mice in Fer-1 and DSS + Fer-1 groups were intraperitoneally injected with 1 mg/kg Fer-1 (GC10380, GLPBIO, Montclair, CA, USA) 7 times a week before DSS induction. Mice in Con group were allowed to drink water at the same time. Stool consistency, mouse weights, and presence of macroscopic fecal blood were recorded at daily sacrifice [23].

### 2.2. Cell Culture and Drug Treatment

HCT116 cells (kindly provided by Dr. Dingding from China Medical University) were cultured in HCT116 Complete medium (CM-0096, Procell Life Science & Technology Co., Ltd., Wuhan, China) at 37 °C in a humidified 5% CO_2_ atmosphere [24]. Cells were randomly divided into six groups: Con, VD (1,25(OH)_2_D_3_), LPS (Sigma-Aldrich, St. Louis, MO, USA), LPS + VD, Fer-1 (GC10380, GLPBIO, USA), and LPS + Fer-1. Cells in LPS group were incubated with 1 μg/mL LPS for 12 h. Cells in VD and LPS + VD groups were treated with 2 × 10^−8^ M 1,25(OH)_2_D_3_ for 24 h. LPS + VD group was first incubated with 1,25(OH)_2_D_3_ (2 × 10^−8^ M), and LPS + Fer-1 group was incubated with Fer-1 (0.1 μM) for 24 h before being treated as LPS group.

All in vitro experiments were conducted in 6-well plates seeded with 2 × 10^6^ cells per well. HCT 116 cells were collected after all the experimental procedures were finished.

### 2.3. ACSL4 Transfection in HCT116

We cloned the full length of the ACSL4 gene and inserted it into the pcDNA3.1 vector to construct plasmids overexpressing ACSL4. Primers are as follows, respectively:

F, 5′-CGGGGTACCAAGCACCATTTTAGAAGCCTTTCC-3′

R, 5′-AAAGCGGCCGCTTATTTGCCCCCATACATTCGT-3′

We used Lipofectamine 3000 (Invitrogen, Carlsbad, CA, USA) to transfect HCT116 cells with ACSL4 plasmid or vector, following the instructions. Western blot and RT-PCR were used to assess the transfection efficiency of ACSL4. Experimental groups included ACSL4^+/+^ Con, ACSL4^+/+^ VD, ACSL4^+/+^ LPS, and ACSL4^+/+^ LPS + VD.

### 2.4. Assessment of Severity of Colitis

The Disease Activity Index (DAI) is a comprehensive score based on the samples’ weight loss percentage, stool consistency, and gross bleeding, which can reflect the degree of damage to colon intestinal tissue well [25]. The specific scores of DAI were tested according to the previously established scoring system evaluation in Table 1.

### 2.5. Evaluation of the Colon Macroscopic Damage Index (CMDI)

CMDI scores, which are frequently used to assess colon injury, were assessed according to the criteria described in Table 2 [26].

### 2.6. Histopathology and Immunohistochemistry (IHC)

The colon tissue of the mice was harvested, and the damage of intestinal mucosa was quantified by HE staining with histological score [27]. All the colon tissues were fixed overnight in 4% paraformaldehyde, embedded in paraffin. The “Swiss rolls” method was used to prepare the sections of mouse colons with 4 μm slices. The sum of the epithelial damage and inflammatory cell infiltration scores was represented by the histological score. For immunohistochemistry staining, sections of paraffin-embedded tissues were dewaxed and heated in a pressure cooker. Endogenous peroxidase activity was blocked by using H_2_O_2_ [28]. Sections were then incubated with anti-ACSL4 (Santa Cruz, Santa Cruz, CA, USA, 365230), anti-GPX4 (Abmart, Shanghai, China, TD6701), anti-COX-2 (Wanleibio, Shenyang, China, WL01750), and anti-IL-6 (Wanleibio, WL03074) followed by secondary antibody and diaminobenzidine (DAB) treatments. All the images were acquired with a Leica 2000 microscope.

### 2.7. Western Blot Analysis

Proteins of cells and tissue were sonicated in RIPA buffer (Beyotime Co., Nantong, Jiangsu, China). Sodium dodecyl sulfate polyacrylamide gel electrophoresis (SDS-PAGE) with 50 mg protein was performed and then transferred to PVDF membranesImageJsoftware (Java 1.6.0_24, National Institutes of Health, Bethesda, MD, USA) was used to analyze the protein bands. Primary antibodies included the following: ACSL4 (Santa Cruz, 365230), GPX4 (Abmart, TD6701), SLC7A11 (Proteintech 26864-1-AP, Rosemont, IL, USA), GAPDH (Proteintech 60004-1-Ig), COX-2 (Wanleibio, WL01750), SOD1 (Wanleibio, WL01846), and IL-6 (Wanleibio, WL03074). Secondary antibodies included the following: HRP Conjugated Goat Anti-Mouse (Proteintech SA00001-1) or Anti-Rabbit IgG (Proteintech SA00001-2).

### 2.8. Real-Time PCR

Trizol reagent (Invitrogen, USA) was used to extract total RNA, and then RNA was converted to cDNA using the Hiscript III RT SuperMix for qPCR (R323-01; Vazyme Biotech, Nanjing, China). The obtained cDNA was detected by quantitative PCR using the QuantiTect SYBR Green I Kit (Q711-02/03; Vazyme). All these data were analyzed by Ct ^(2−(ΔCt−Cc))^ method, with GAPDH as the housekeeping gene. All the sequences of primers are shown in Table 3.

### 2.9. Measurement of Glutathione (GSH) and Malondialdehyde (MDA) in Colonic Tissues and Cells

Fresh colon tissues (50 mg) and HCT116 cells (2 × 10^6^) were homogenized with phosphate-buffered saline (PBS). Supernatant was collected after centrifugation [29]. The oxidative and anti-oxidative stress markers in tissue and cells were examined, including GSH and MDA. GSH assay kits (WLA104, Wanleibio, China) and MDA assay kits (WLA048a, Wanleibio, China) were used following the manufacturer’s instructions.

### 2.10. Measurement of Myeloperoxidase (MPO)

Tissue samples and cells were prepared as described above. MPO, an important marker for inflammatory damage, was expressed as units per gram tissue (U gram tissue^−1^) and taking 1 unit of the enzyme associated with a decrease of 1 mol of hydrogen peroxide (H_2_O_2_) per minute [30]. MPO assay kit (A044-1-1, Nanjingjiancheng, Nanjing, China) was used following the manufacturer’s instructions.

### 2.11. Iron Measurements

Tissue samples and HCT116 cells were prepared as described above. The level of Iron was measured by using Iron Assay Kit (A039-2-1, Nanjingjiancheng, China), following the manufacturer’s instructions.

### 2.12. Statistical Analysis

GraphPad Prism version 8 software (San Diego, CA, USA) was used to analyze statistical differences between groups. The two-way analysis of variance (ANOVA) and student’s *t*-test followed by Bonferroni test were used to evaluate various groups. A *p* value of <0.05 was considered statistically significant.

## 3. Results

### 3.1. VD/VDR Attenuated DSS-Induced UC in Mice

To establish the classic acute UC model, wild ICR mice were treated with 3% DSS drinking water for 10 days. Compared with the Con group, the administration of CCE alone increased the colon length slightly (Figure 1A). In the DSS and DSS + VD groups, we found that the colon length (Figure 1A) and weight (Figure 1B) of mice decreased markedly; however, these phenomena were notably improved in the DSS + VD group, which was pretreated by CCE for two weeks. From the third day, the DAI scores of mice in the DSS and DSS + VD groups began to rise (Figure 1C); nevertheless, from the sixth day, the scores in the DSS + VD group tended to be stable. The score of CMDI is frequently used to assess the damage to colon tissue. The experimental results showed that the CMDI (Figure 1D) score of the DSS group is much higher than that of the Con and VD groups. Although there was no statistical difference in the scores between the Con and VD groups, the scores of the DSS + VD group were lower than that of the DSS group, indicating that, to some extent, CCE pretreatment could protect colon tissue from pathological damage induced by DSS. We observed that there existed extensive mucosal ulcers, reduced crypts, epithelial edema, and inflammatory cell infiltrations in DSS mice by HE staining (Figure 1E). After CCE pretreatment, these histological damages were alleviated in the DSS + VD group. In addition, we also verified the role of VD by detecting other oxidative stress indicators (including MDA and MPO). The accumulation of MDA and MPO usually represented inflammatory damage. The contents of them in the DSS group were much higher than those in the Con and VD groups. Similarly, compared with the DSS group, the contents of the two indicators decreased, and data were statistically significant in the DSS + VD group. All these results indicate that VD can partly protect colon tissue from DSS-induced inflammation and pathological damage. Western blot and Immunohistochemistry were then used to detect the inflammatory damage at the protein level. The expression of VDR in the VD group and DSS + VD group increased remarkably, as observed by Western blot (Figure 1G) and Immunohistochemistry (Figure 1F). The experimental results showed that in comparison with the DSS group, the expression levels of COX-2 and IL-6 (both were inflammatory factors) in colon tissue decreased significantly after CCE pretreatment in the DSS + VD group. SOD1, an enzyme that plays an important role in protecting tissue against oxidative stress, is indispensable in suppressing the development of colitis [31]. In the DSS group, the expression of SOD1 decreased significantly, but it increased slightly in the DSS + VD group. In general, our data confirmed that the VD/VDR can rescue inflammation and oxidative stress injury in DSS mice.

### 3.2. VD/VDR Inhibits UC Inflammation In Vitro

To further elucidate the inflammatory injury, we used LPS to simulate the UC model in HCT 116 cells. Whether for MDA or MPO, the contents in the model group were much higher than those in the Con and VD groups, suggesting that there existed an obvious oxidative stress damage in the cell model; meanwhile, when pretreated with 1,25(OH)_2_D_3_, the contents of both indicators in LPS + VD group decreased and had a statistical significance between the LPS and LPS + VD groups. At the mRNA (Figure 2B) and Western blot (Figure 2D) levels, the 1,25(OH)_2_D_3_ in VD and LPS + VD groups can conspicuously increase the expression of VD by activating VDR. The expression of COX-2 in the LPS group was significantly higher than that in the Con and VD groups, while the content in the LPS + VD group was lower than that in the LPS group, with statistical significance (Figure 2B,D). The trend of IL-6 expression was similar to that of COX-2. IL-6 expression in the model group was much higher than any other group, while for VD pretreated, its expression in the LPS + VD group was lower than that in the LPS group (Figure 2D). On the contrary, the level of SOD1 decreased in the LPS model but slightly improved in the LPS + VD group (Figure 2B,D). All these indicated that 1,25(OH)_2_D_3_ can reduce the inflammatory reaction of UC in vitro through the VD/VDR pathway.

### 3.3. Ferroptosis Was Involved in the Pathological Process of UC

Ferroptosis was found to be a new pattern of cell death, mainly caused by Iron-dependent lipid peroxidation, which can be partially inhibited by Fer-1. The colon length (Figure 3A) and body weight (Figure 3B) exhibited a dramatic drop in the 3% DSS group compared with the Con group. Western blot and RT-PCR were employed to monitor whether ferroptosis was involved in the pathological process of UC. The levels of ACSL4 and GPX4 (both were considered specific markers of ferroptosis) showed a completely opposite trend. ACSL4 was significantly elevated, while GPX4 was decreased in DSS-induced mice in comparison with the Con group. Meanwhile, Iron content in UC mice increased, and GSH declined compared to the Con group. All of these suggested the accumulation of Iron and lipid peroxidation take part in the process of UC. We then intraperitoneally administered Fer-1 to mice to further clarify the important role of the ferroptosis process in UC. Colon length (Figure 3A) and body weight (Figure 3B) were slightly improved in the DSS+Fer-1 group than in the DSS group. Meanwhile, the DAI (Figure 3C) and CMDI (Figure 3D) scores used to evaluate the pathological damage were also decreased by Fer-1 in the DSS+Fer-1 group, compared with the DSS group. Not surprisingly, after pretreatment with Fer-1, the level of Iron (Figure 3E) in the DSS+Fer-1 group was lower than in the DSS group, while obviously, more GSH (Figure 3F) accumulated in the DSS+Fer-1 group than in the DSS group [32]. The expression of ACSL4 in the DSS group was higher than in the DSS+Fer-1 group, while GPX4 in the DSS group was lower than in the DSS+Fer-1 group (Figure 3G,H). Cell experiments also confirmed this tendency (Figure 3J). In comparison with the LPS group, the content of Iron (Figure 3I) decreased, and GSH (Figure 3K) increased in the LPS+Fer-1 group in cell cultures. In summary, our data demonstrated that ferroptosis played an indispensable role in the UC’s pathological process, and Fer-1 can attenuate the pathological process of UC.

### 3.4. VD/VDR Regulated the Expression of ACSL4 and Other Ferroptosis Relative Proteins In Vivo and In Vitro

Not unexpectedly, Iron content (Figure 4A,E) was found to have accumulated while GSH content declined in UC models (Figure 4B,F). After administration of VD, the Iron content (Figure 4A) in DSS + VD was less than that in the DSS group, while the GSH content (Figure 4B) was increased. The LPS-induced cell model also confirms this point (Figure 4E,F). Then, we assessed the effect of CCE on DSS-induced ferroptotic phenotype. Results showed that pretreatment with CCE can notably decline the expression of ACSL4 and enhance the expression of GPX4 in the DSS + VD group compared with the DSS group, and the data were statistically significant (Figure 4C). As we predicted, cell experiments have shown a similar tendency (Figure 4D,G). In UC models, we observed that the expression of SLC7A11 apparently declined more than those in the Con and VD groups, demonstrating ferroptosis was involved in UC’s pathological process (Figure 4C,G). The results, however, were not statistically significant between the DSS group and the DSS + VD group (Figure 4C,G). In general, our results may predict that VD/VDR can downregulate ACSL4, upregulate GPX4, and change any other relative indicators in UC.

### 3.5. ACSL4 Plays an Indispensable Role in Vitamin D Mitigation of UC

To verify whether VD alleviates UC through ferroptosis by downregulating ACSL4, the ACSL4 gene was overexpressed in HCT116 cells. The significant increase in ACSL4 protein (Figure 5A) and mRNA (Figure 5C) in the ACSL4^+/+^ group compared with the vector group was considered a successful sign of overexpression. We found that the contents of MDA (Figure 5B) and MPO (Figure 5E) in the ACSL4^+/+^ group were much higher than those in the vector. Both MDA and MPO contents in the model group in the vector were significantly higher than those in the Con and VD groups, while the contents of MDA and MPO in the LPS + VD group were lower than those in the LPS group, with statistical significance. Western blot results of the cell model showed (Figure 5D) that in the vector, compared with the LPS group, the expression levels of ACSL4, COX-2, and IL-6 in the LPS + VD group decreased, while the expression levels of GPX4 and SOD1 increased. However, these phenomena were not significant in the ACSL4^+/+^ group, and there was no statistical significance. This indicates that VD’s ability to reduce ferroptosis in inflammation is limited by the overexpression of ACSL4 (Figure 5D). In conclusion, these results suggested that the overexpression of ACSL4 may partially undermine the protective effect of VD on UC.

## 4. Discussion

UC is a nonspecific inflammatory disease of colon and rectum tissue with unclear and complex etiology. In our previous study, we confirmed that CCE can attenuate the inflammation of UC in mice by downregulating COX-2 and IL-6 [20]. To elucidate the role of VD in the pathological process of UC, we used the classic models of DSS-induced mice and LPS-induced cells as before. Consistent with previous studies, our experiment proved that VD can improve colon length and body weight and decrease the scores of DAI and CMDI in the DSS + VD group compared with the DSS group. HE staining showed that the disordered colon structures were relieved by CCE. All of these demonstrate that additional administration of VD can alleviate the progression of UC. In addition, the levels of proteins suggested that VD can decrease COX-2 and IL-6 to suppress inflammation. As for immune-regulatory factors, oxidative stress has been considered a crucial mechanism in the pathological process of UC [33]. Therefore, we tested the level of MDA and MPO in tissue; both were downregulated by VD in vivo [34,35]. Then, HCT116 cells were used in subsequent studies to verify the inflammation injury in vitro. Cell experiments performed the same tendency in accordance with the mouse model, demonstrating that VD can inhibit inflammation and oxidative stress injury in UC.

Ferroptosis is a new programmed cell death mode that depends on a series of positive-acting enzymatic reactions. It includes the biosynthesis of PUFA-containing phospholipids, which are the substrates of pro-ferroptotic lipid peroxidation products by ACSL4 and LPCAT3 [36]. The main mechanism of ferroptosis is lipid peroxidation, which induces cell death. Compared with the Con group, DSS mice showed colon length decrease, body weight loss, and raised DAI and CMDI scores. Meanwhile, the expression of the antioxidant system GSH and GPX4 was decreased [11]. In our study, the accumulation of Iron and the decline of GSH were found in the UC model. All of these results demonstrate that ferroptosis takes part in the process of UC. To further verify the important role of ferroptosis in UC, we intraperitoneally injected mice with Fer-1 for seven days before establishing the DSS model [37]. Consistent with previous research, the colon length shortening and body weight decline in the DSS+Fer-1 group were relieved [5,17,38] compared with the DSS group. As expected, DSS+Fer-1 mice showed lower scores of DAI and CMDI than those of the DSS group. The symbol expression of ferroptosis biomarker ACSL4 was decreased, and GPX4 was increased by Fer-1 in the DSS+Fer-1 group, compared with the DSS group. All of our observations suggested that Fer-1 could alleviate ferroptosis.

VD/VDR, combined with 1,25(OH)_2_D_3_ to achieve its biological functions [39], has been proved in correlation with ferroptosis in cisplatin-induced AKI through the transcriptional regulation of the GPX4 gene [40]. However, whether VD can alleviate UC through the process of ferroptosis has not been confirmed. In our study, we first measured the levels of Iron and GSH; as we expected, VD pretreatment can reduce Iron content and increase the level of GSH. We then proved the expression of ACSL4 was slightly decreased while GPX4 increased in DSS + VD and LPS + VD groups than in models. SLC7A11, a protein exporting intracellular glutamate in exchange for extracellular cystine in a gradient-dependent manner, belongs to the cystine/glutamate exchanger (XcT). The level of SLC7A11 notably declined in VD-deficient mice compared with the normal group, indicating that VD deficiency may downregulate the expressions of XcT through targeting VDR in Alzheimer’s disease [41]. However, in our UC experiment, the level of SLC7A11 was only lower in the DSS and DSS + VD groups compared to the Con and VD groups but did not present a statistical significance between the DSS and DSS + VD groups. We speculated that the main reason for the above result is that the sample size is too small or the protein is less expressed in the colon. The specific reasons need to be further studied. According to these results, we predicted that the inflammation and oxidative stress injury would be suppressed by VD.

In order to further verify the function of ACSL4, the gene was successfully overexpressed by using pcDNA3.1 in vitro. By comparing the protein expression of group ACSL4^+/+^LPS and ACSL4^+/+^LPS + VD, we found that after ACSL4 was overexpressed, the expressions of COX-2 and IL-6 were not significantly inhibited by VD pretreatment, while SOD1 and GPX4 were significantly restricted. There was no significant difference between the ACSL4^+/+^LPS and ACSL4^+/+^LPS + VD groups, indicating that a high level of ACSL4 may inhibit the relieving effect of VD on ferroptosis to some degree. In the vector group, the contents of MDA and MPO in the UC model were significantly higher than in the other three groups, while both indicators in the ACSL4^+/+^LPS and ACSL4^+/+^LPS + VD groups did not present a statistical difference, demonstrating that the overexpression of ACSL4 may partially inhibit the mitigating effect of VD on oxidative stress injury in the UC model. Our experimental results showed that VD can alleviate inflammation and oxidative stress damage while reducing the expression of ferroptosis-related proteins in UC models. However, after the ACSL4 gene was overexpressed, the above effects were significantly inhibited. Therefore, we speculated that VD may alleviate the pathological damage of UC partly by downregulating the ACSL4 gene.

In summary, our research elucidated that ferroptosis takes part in the pathological process of UC as an indispensable role; meanwhile, VD contributes to mitigating ferroptosis at least partially via downregulating ACSL4 protein. However, the research also has some limitations. The specific mechanism of VD regulating ACSL4 as a transcription factor remains to be proved. Further experiments are encouraged and needed to verify the in-depth molecular mechanism in clinical application.

## Figures and Tables

**Figure 1 nutrients-15-04845-f001:**
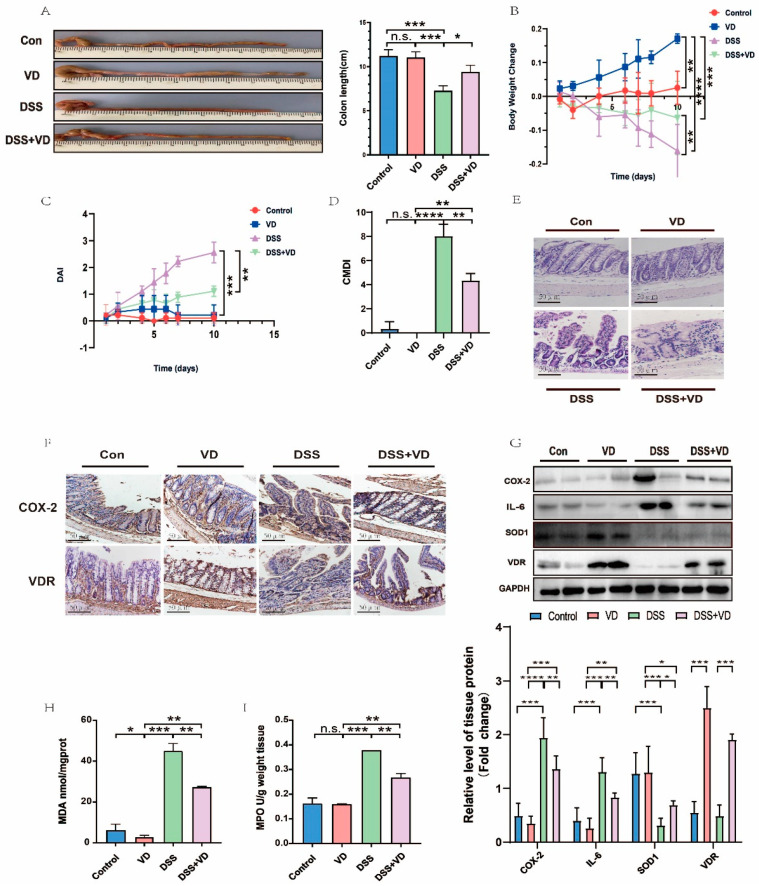
VD/VDR attenuated DSS-induced UC in mouse model. (**A**) The length of colon was shortened, thickened, and congested in DSS group, in comparison with DSS + VD group; (**B**) body weight declined in DSS group compared with DSS + VD group; (**C**,**D**) DAI and CMDI scores were notably higher in DSS group than in DSS + VD group; and (**E**) HE staining of colon was performed after mice were sacrificed. Severe and extensive inflammatory cell infiltration was shown in DSS and DSS + VD groups (scale bar = 50 μm). (**F**) The expression and distribution of COX-2 and VDR with immunohistochemical stain were shown (scale bar = 50 μm). DSS group showed high expression of COX-2 in comparison with DSS + VD group. (**G**) The changes in protein levels in different groups were analyzed by Western blot. The contents of COX-2 and IL-6 in DSS group were obviously augmented, whereas the expression of SOD1 was decreased compared with DSS + VD group. (**H**,**I**) The levels of MDA and MPO in DSS group were higher than in DSS + VD group. Values shown are means ± SD. * *p* < 0.05, ** *p* < 0.01, *** *p* < 0.001, **** *p* < 0.0001, and n.s., not significant. (n = 3 in each group).

**Figure 2 nutrients-15-04845-f002:**
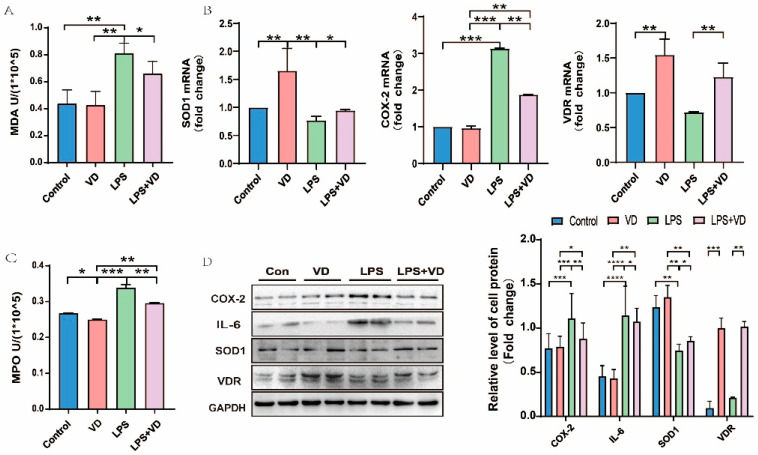
The inflammation and oxidative stress factors in UC were suppressed by VD/VDR. (**A**) LPS group showed high level of MPO compared with LPS + VD group. (**B**) RT-PCR was used to detect the mRNA levels of SOD1, COX-2, and VDR. (**C**) In comparison with LPS group, MDA content in LPS + VD group was significantly decreased. (**D**) Western blot showed inflammatory factors and oxidative stress proteins in LPS-induced cell model. Values shown are means ± SD. * *p* < 0.05, ** *p* < 0.01, *** *p* < 0.001 and **** *p* < 0.0001. (n = 3 in each group).

**Figure 3 nutrients-15-04845-f003:**
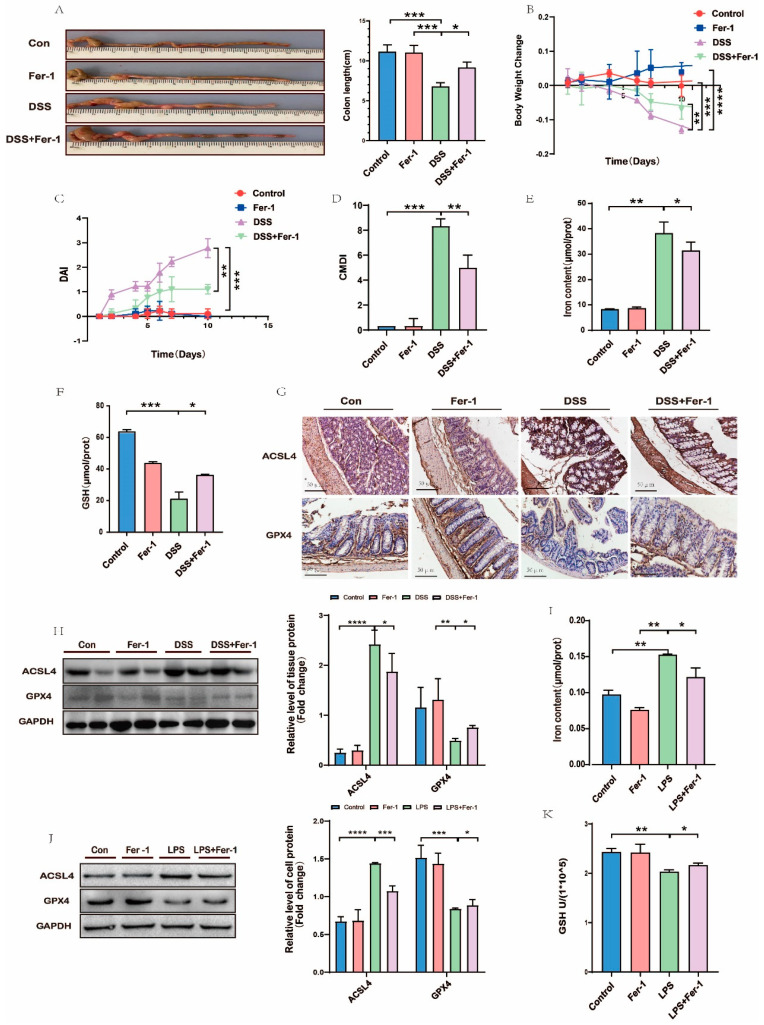
Ferroptosis took part in the pathological process of UC. (**A**) The length of colon was shortened, thickened, and congested in DSS group, compared with DSS + Fer-1 group; (**B**) body weight was lower in DSS group than in DSS + Fer-1; (**C**,**D**) DAI and CMDI scores were significantly higher in DSS group than in DSS + Fer-1 group; and (**E**,**I**) Iron content in UC models was higher than in DSS + Fer-1 and LPS + Fer-1 groups. (**F**,**K**) GSH content in UC models was less than in DSS + Fer-1 and LPS + Fer-1 groups. (**G**,**H**,**J**) In protein levels, ACSL4 was higher in UC models than in DSS + Fer-1 and LPS + Fer-1 groups, and GPX4 showed an opposite tendency (scale bar = 50 μm). Values shown are means ± SD. * *p* < 0.05, ** *p* < 0.01, *** *p* < 0.001 and **** *p* < 0.0001. (n = 3 in each group).

**Figure 4 nutrients-15-04845-f004:**
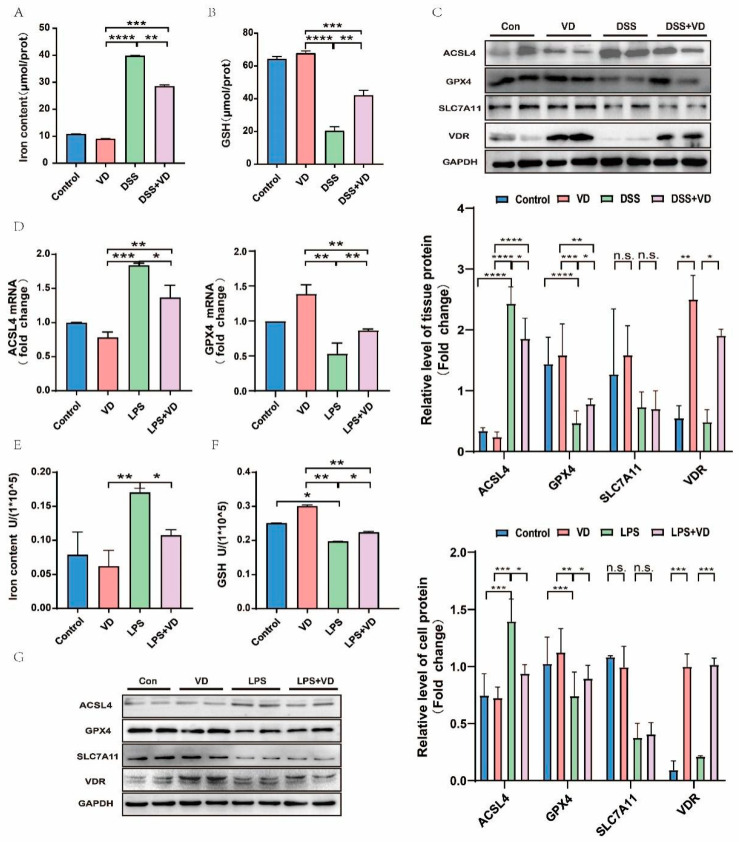
ACSL4 and other ferroptosis-relative proteins were regulated by VD/VDR. (**A**,**E**) The level of Iron in UC models was higher than in VD pretreatment groups. (**B**,**F**) The GSH level was increased in DSS + VD and LPS + VD groups, compared with DSS and LPS groups. (**C**,**G**) The expression of relative ferroptosis proteins is shown in vivo and in vitro. (**D**) The results of RT-PCR showed the change in ACSL4 and GPX4 in cells. Pretreated with VD, the level of ACSL4 slightly declined, and GPX4 improved in LPS + VD group compared with LPS group. Values shown are means ± SD. * *p* < 0.05, ** *p* < 0.01, *** *p* < 0.001 and **** *p* < 0.0001, n.s., not significant.(n = 3 in each group).

**Figure 5 nutrients-15-04845-f005:**
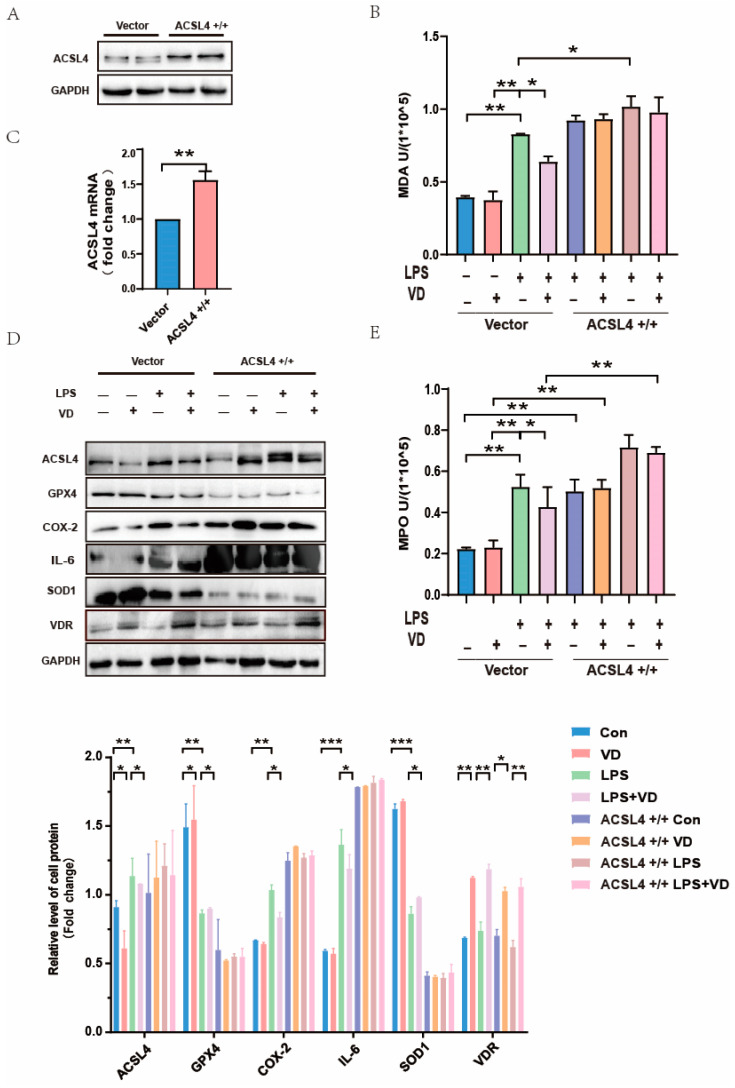
The crucial role of ACSL4 in VD mitigation of UC. (**A**,**C**) Western blot and RT-PCR showed that ACSL4 gene was overexpressed in HCT116 cells. (**B**,**E**) The contents of MDA and MPO attenuated after giving VD, but this tendency abated after the transfection of ACSL4^+/+^. (**D**) Expression levels of ACSL4, GPX4, inflammation indicators (COX-2, IL-6), and SOD1 after cells were transfected with ACSL4^+/+^ plasmids by using Western blot. Values shown are means ± SD. * *p* < 0.05, ** *p* < 0.01, and *** *p* < 0.001. (n = 3 in each group).

**Table 1 nutrients-15-04845-t001:** DAI criteria.

Grade	Weight Loss (%)	Stool Consistency	Gross Bleeding
0	0	Normal	N/A
1	1–5	Mild soft	—
2	5–10	Soft and wet	Hemoccult positive
3	10–20	Half-loose stool	—
4	>20	Loose stool	Gross bleeding

ACSL4, GPX4, inflammation indicators (COX-2, IL-6), and SOD1 after cells were transfected with ACSL4^+/+^ plasmids by using Western blot.

**Table 2 nutrients-15-04845-t002:** CMDI criteria.

Score	Criteria
0	No damage
1	Hyperemia
2	Hyperemia and thickening without ulceration
3	Ulceration at a single site
4	Two or more sites of ulceration or inflammation
5	Ulceration or inflammation extending > 1 cm along the length
6–10	Damage covering > 2 cm along the length of colon, with the score being increased by 1 for each additional cm of involvement

**Table 3 nutrients-15-04845-t003:** Sequences of the primers used for RT-PCR.

Gene	Mouse	Human
*ACSL4*	F: TCCTTCGTGACTACTGCCGAG R: GTTATAGGTGGTTTCGTGGAT	CATCCCTGGAGCAGATACTCT TCACTTAGGATTTCCCTGGTC
*VDR*	F: CAAGGACAACCGACGCCACTG R: CCTCCTCCTCCTTCCGCTTCAG	CTGGTGACTTTGACCGGAAT CTGCACCTCCTCATCTGTGA
*COX-2*	F: GGTTGCTGGTGGTAGGAATC R: TAAAGCGTTTGGGGTACTCA	CTAGAGCCCTTCCTCCTGCG GCTGGGCAAAGAATGCAAT
*SOD1*	F: CTCAGGAGACCATTGCATCA R: ACAAGCCAAACGACTTCCAG	TGTCATGCAGTCCCTTGGAT CATTCTCGCCCTGGATCTCT
*GPX4*	F: GCCCCTCCATCTACGACTTC R: TTGGTGATGATGCAGACGAAC	GGAGCCAGGGAGTAACGAAG GACGGTGTCCAAACTTGGTG
*GAPDH*	F: AAATCAAGTGGGGCGATGCT R: TGGTTCACACCCATGACGAA	ACCCAGAAGACTGTGGATGG TCAGCTCAGGGATGACCTTG

## Data Availability

The data that support the findings of this study are available from the corresponding author upon reasonable request.

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
