# Peer review of "Vitamin D Attenuates Ulcerative Colitis by Inhibiting ACSL4-Mediated Ferroptosis"

_nutrients, 2023, doi:10.3390/nu15224845_

Round 1

Reviewer 1 Report

Comments and Suggestions for Authors

In the manuscript “Vitamin D attenuates ulcerative colitis by inhibiting ACSL4-2 mediated ferroptosis”, Shuo Gao and collaborators studied the role of vitamin D in the treatment of ulcerative colitis using various types of techniques and different conditions. They conclude that vitamin D plays a role in attenuating ulcerative colitis by inhibiting Acyl-CoA synthetase long-chain family member 4 implicated in ferroptosis. Although the work is very interesting it needs to be revised a lot. My opinion is minor revision.

Below is a list of the main indications:

·       Improve your English

·       In general, the writing is difficult to follow, there are paragraphs that should be moved to different positions. For example, there are protocol descriptions in the results.

·       The paper is full of acronyms, including acronyms that are not defined; add a list of acronyms with the meaning.

·       Improve the introduction by introducing more references and better specifying the purpose of the work

·       Specify the system Xc activity

·       Specify the role of the sugars used or give precise bibliographical information

·       Reference is made to a previous work by the authors for which no information is given

·       Improve the discussion

·       Materials and Methods: the total number of mice used, the number of mice used for each type of experiment and how many replicates were made are not indicated

·       Define “male ICR mice”

·       Materials and Methods: indicate the dilution of the antibodies used in immunohistochemistry, the manufacturing company and the code

·       In Materials and Methods the CCE and 1,25(OH)D3 treatments are not specified, insert at least one reference and specify the reason for use

·       The Figs need to be organized better as there are not all the necessary indications and the groupings within the same figure are not immediately clear. Furthermore, the letters of the figures are too far from the image to which they refer and are difficult to see

·       The legends do not faithfully reflect what is shown in the figure

·       In general, there is a lack of information on histology figures

·       Why are samples always loaded twice in western blots?

·       Standardize the typology of graphs

·       How DAI and CMDI are calculated?

·       Fig. 1: The fold change of protein tissue level graph is not discussed

·       Fig. 1E CCE treated samples are not shown

·       Fig. 1F: VDR is not shown. Difficult to interpret the data

·       Fig. 1 G and corresponding graph: vitamin D when administered with DSS does not recover inflammation (Il6 and COX2)

·       Fig. 1G: the trend of VDR is not described

·       Fig. 2: some graphs are not indicated in the legend (COX2, VDR) and insert the letters on all images

·       Fig. 4G: the fold change of protein level does not correspond to the intensity of the protein bands in the western blot

·       Uniform the bibliography

Comments on the Quality of English Language

The English language can be improved

Author Response

Response to Reviewer 1 Comments

Dear expert,

Thank you for taking the time to give your comments, we have made changes according to your suggestions, if there are still problems, please enlighten us.

Point 1:

  • Improve your English
  • In general, the writing is difficult to follow, there are paragraphs that should be moved to different positions. For example, there are protocol descriptions in the results.
  • The paper is full of acronyms, including acronyms that are not defined; add a list of acronyms with the meaning.

Response 1: Thank you very much for your suggestion, we have modified it. Meanwhile, Native English-speakers have reviewed and edited the text, and we will upload the latest edition as soon as possible. Also, we have added a list of acronyms with the meaning, on page 14-15, lines 435-468 in the text.

1,25-dihydroxyvitamin D3 (1,25(OH)2D3)

Acyl-CoA synthetase long-chain family member 4 (ACSL4)

Arachidonic acid (AA)

Cholecalciferol cholesterol emulsion (CCE)

Colon Macroscopic Damage Index (CMDI)

Control (Con)

Crohn’s disease (CD)

Cyclooxygenase-2 (COX-2)

Dextran Sulfate Sodium Salt (DSS)

Diaminobenzidine (DAB)

Disease Activity Index (DAI)

Ferrostatin-1 (Fer-1)

Gastrointestinal tract (GIT)

Glutathione (GSH)

Hydrogen peroxide (H2O2)

Immunohistochemistry (IHC)

Inflammatory bowel disease (IBD)

Interleukin- 6 (IL-6)

Lipopolysaccharide (LPS)

Long chain acyl coenzyme A synthase family (ACSLs)

Malondialdehyde (MDA)

Myeloperoxidase (MPO)

Phate-buffered saline (PBS)

Polyunsaturated fatty acid (PUFAs)

Recombinant Glutathione Peroxidase 4 (GPX4)

Recombinant Solute Carrier Family 7, Member 11 (SLC7A11)

Reverse transcription-polymerase chain reaction (RT-PCR)

Sodium dodecyl sulfate polyacrylamide gel electrophoresis (SDS-PAGE)

Superoxide Dismutase 1 (SOD1)

Ulcerative Colitis (UC)

Vitamin D (VD)

Vitamin D receptor (VDR)

Xc−activity (XcT)

Point 2: Improve the discussion

Response 2: Thank you for your patience and careful suggestion. We have rechecked and made modifications.

Point 3: Materials and Methods: the total number of mice used, the number of mice used for each type of experiment and how many replicates were made are not indicated

Response 3: Thank you for your suggestion. We have added it on page 3, lines 97-99 in the text: 30 mice (n = 5 per group) were randomly divided into the following 6 groups, including Control (Con), VD, Fer-1, DSS, DSS+VD and DSS+Fer-1 groups. We performed a total of three replicate experiments.

Point 4: Define “male ICR mice”

Response 4: ICR mouse is an international closed group mouse. The Institute of Cancer Research in the United States sent the mice to various countries for breeding experiments, which are called ICR mice. We used male mice of this strain.

Point 5: Materials and Methods: indicate the dilution of the antibodies used in immunohistochemistry, the manufacturing company and the code

Response 5: Thank you for your suggestion. We have added it on page 4, lines 153-155 in the text: anti-ACSL4(Santa Cruz, 365230), anti-GPX4(Abmart, TD6701), anti-COX-2(Wanleibio, WL01750) and anti-IL-6 (Wanleibio, WL03074).

Point 6: In Materials and Methods the CCE and 1,25(OH)D3 treatments are not specified, insert at least one reference and specify the reason for use

Response 6 : Thank you very much for your suggestion!

VD and DSS+VD groups were given CCE (10 μl in 100 ml of water) [21] , a VD analog for 2 weeks[23].( Line 103-104)

In our previous research, we found that 2 X 10-8 M vitamin D could inhibit the migration and proliferation of H460 cells, and promote apoptosis (Liu, Ning., Li, Xiaofeng., Li, Xiaofeng., Fu, Yu.,  & Li, Ye.. (2020). Inhibition of lung cancer by vitamin D depends on downregulation of histidine-rich calcium-binding protein. Journal of advanced research, 29). So we firstly choose this concentration to induce cells. (Line 115-116)

Point 7: The Figs need to be organized better as there are not all the necessary indications and the groupings within the same figure are not immediately clear. Furthermore, the letters of the figures are too far from the image to which they refer and are difficult to see

  • The legends do not faithfully reflect what is shown in the figure

Response : Thank you for your suggestion. We have readjusted the figures and upload the latest edition as soon as possible!

  • In general, there is a lack of information on histology figures

Response : Thank you for your suggestion. We have readjusted the figures and upload the latest edition as soon as possible!

  • Why are samples always loaded twice in western blots?

Response : Because we have always selected the two most representative samples from both cells and tissues to explain the results, so the samples are always loaded twice in Western blots.

  • Standardize the typology of graphs

Response : Thank you very much for your suggestion! We will upload the latest edition as soon as possible!

  • How DAI and CMDI are calculated?

Response : The disease activity index (DAI) was combined with the percentage of weight loss, stool consistency and gross bleeding. The total score of the three results was divided by 3 to obtain the DAI value. Namely, DAI=(body mass index+stool consistency+gross bleeding)/3.

The CMDI score is based on visual observation of colonic mucosal damage, including the degrees of mucosal congestion, edema, and the presence or absence of ulcers, erosion, etc.

  • Fig. 1: The fold change of protein tissue level graph is not discussed

  • Fig. 1E CCE treated samples are not shown

  • Fig. 1F: VDR is not shown. Difficult to interpret the data

  • Fig. 1 G and corresponding graph: vitamin D when administered with DSS does not recover inflammation (Il6 and COX2)

  • Fig. 1G: the trend of VDR is not described

  • Fig. 2: some graphs are not indicated in the legend (COX2, VDR) and insert the letters on all images
  •  
  • Fig. 4G: the fold change of protein level does not correspond to the intensity of the protein bands in the western blot

Response : Thank you very much for your suggestion! We feel really sorry for we may reorganize the figures. We will upload the latest edition as soon as possible!

Point 8:  Uniform the bibliography

Response 8 : Thank you very much for your suggestion! We will upload the latest edition as soon as possible!

Thank you again for your prompt reply! We have improved the article in English, which may take a few days. At the same time, we have readjusted the figures. We guarantee to upload the latest version of the article as soon as possible! Thank you very much for your support and understanding! Looking forward to hearing from you anytime!

Thanks again

Sincerely yours,

Juan Kong

Reviewer 2 Report

Comments and Suggestions for Authors

 Please correct of clarify the following points: 

L27: “threatens” instead of “threaten”

L43: add … diseases are still..

L71: “processes” instead of “process”

L78: In our previous study, we have confirmed…

L96: ..  intraperitonealy injected with 1mg/kg Fer-1

L99: sacrifice

L136: The colorecteal…. Of the mice: I think there is a word missing

L138: …method was ..

L189: did not increase or decrease? Check with Fig 1A

L191: .. group which was pretreated..

L200: delete an

L204: .. MPO usually represent …

L206: .. and data were statistically ..

L210: .. remarkably, as observed..

L223: ..was preformed after ..

L 243: group instead of groups

L263-4: GSH declined compared to ..

L265: ..peroxidation take part ..

L270: ..after pretreatment..

L271: ..was lower than..

L276: .. in cell cultures.

L291: content declined in UC models … After administration of VD..

L294: pretreatment..

L295: enhance instead of enhanced

L361: ..ferroptosis takes part..

L366: ..performed low scores ..

L367: … biomarker ACSL4 was decreased..

L392: ..indicating that high level..

L401: that VD may alleviate ..

Comments on the Quality of English Language

Some of the points mentioned in the previous section address to the quality of the english language.

Author Response

Response to Reviewer 2 Comments

Dear expert,

Thank you for taking the time to give your comments, we have made changes according to your suggestions, if there are still problems, please enlighten us.

Point 1:

L27: “threatens” instead of “threaten”

Response: Inflammatory bowel disease (IBD) is a chronic disease[1], seriously threatens people’s health and severely impacts on economic development worldwide.

L43: add … diseases are still..

Response: However, the immunomodulatory effects of VD to improve autoimmune diseases are still unclear[6].

L71: “processes” instead of “process”

Response: PUFAs are involved in a variety of processes, including membrane phospholipid composition, the synthesis of lipid signaling pathways and conduction of ferroptosis signaling of lipid oxidation, and other processes to induce intracellular ferroptosis[19, 20].

L78: In our previous study, we have confirmed…

Response: In the previous study, we have confirmed that VD can effectively alleviate DSS-induced UC via downregulating the expression of pro-inflammatory factors, meanwhile maintaining the integrity of the intestinal mucosal epithelium[21].

L96: ..  intraperitonealy injected with 1mg/kg Fer-1

Response: Mice in Fer-1 and DSS+Fer-1 groups were intraperitonealy injected with 1mg/kg Fer-1 (GC10380, GLPBIO, America) 7 times a week before DSS induction.

L99: sacrifice

Response: Stool consistency, mice weights and presence of macroscopic fecal blood were recorded daily before sacrifice [24]

L136: The colorecteal…. Of the mice: I think there is a word missing

Response: The colorectal colon tissue of the mice were harvested and the damage of intestinal mucosa was quantified by HE staining with histological score[28].

L138: …method was ..

Response: The “Swiss rolls” method was used to prepare the sections of mice colons with 4 μm slices.

L189: did not increase or decrease? Check with Fig 1A

Response: Compared with Con group, the administration of CCE alone did not increased the colon length slightly (Fig. 1A).

L191: .. group which was pretreated..

Response: In DSS and DSS+VD groups, we found that the colon length (Fig. 1A) and weight (Fig. 1B) of mice decreased markedly, however, these phenomena were notably improved in DSS+VD group which was pretreated by CCE for two weeks.

L200: delete an

Response: We observed that there existed an extensive mucosal ulcers, reduced crypts, epithelial edema and inflammatory cell infiltrations in DSS mice by HE staining (Fig. 1E).

L204: .. MPO usually represent …

Response: The accumulation of MDA and MPO are usually represented inflammatory damage. The contents of them in DSS group were much higher than those in Con and VD groups.

L206: .. and data were statistically ..

Response: Similarly, compared with DSS group, the contents of the two indicators decreased and the data was were statistically significant in DSS+VD group.

L210: .. remarkably, as observed..

Response: The expression of VDR in VD group and DSS+VD group increased remarkably, as observed by Western blot (Fig. 1G) and Immunohistochemistry (Fig. 1F).

L223: ..was preformed after ..

Response: (E) HE staining of colon was operated preformed after mice were sacrificed. Severe and extensive inflammatory cells infiltration was shown in DSS and DSS+VD groups. (Scale bar = 50 μm)

L 243: group instead of groups

Response: IL-6 expression in the model group was much higher than any other group, while VD pretreated, the expression of it in LPS+VD group was lower than that in LPS group (Fig. 2D) .

L263-4: GSH declined compared to ..

Response: Meanwhile, iron content in UC mice was increased and GSH was declined compared to with Con group.

L265: ..peroxidation take part ..

Response: All of these suggested the accumulation of iron and lipid peroxidation were take part in the process of UC.

L270: ..after pretreatment..

Response: Not surprisingly, after pretreated pretreatment with Fer-1,

L271: ..was lower than..

Response: the level of Iron (Fig. 3E) in DSS+Fer-1 group was descended lower than in DSS group,

L276: .. in cell cultures.

Response: In comparison with LPS group, the content of Iron (Fig. 3I) decreased and GSH (Fig. 3K) increased in LPS+Fer-1 group in cell cultures.

L291: content declined in UC models … After administration of VD..

Response: Not unexpectedly, Iron content (Fig. 4A, 4E) was found to be accumulated while GSH content was declined in UC models (Fig. 4B, 4F). After giving administration of VD, the Iron content (Fig. 4A) in DSS+VD was less than that in DSS group, while the GSH content (Fig. 4B) was increased.

L294: pretreatment..

Response: Results showed that pretreated pretreatment with CCE can notably decline the expression of ACSL4 and enhance the expression of GPX4 in DSS+VD group compared with DSS group and the data was statistically significant (Fig. 4C).

L295: enhance instead of enhanced

Response: Results showed that pretreated pretreatment with CCE can notably decline the expression of ACSL4 and enhance the expression of GPX4 in DSS+VD group compared with DSS group and the data was statistically significant (Fig. 4C).

L361: ..ferroptosis takes part..

Response: All of these results demonstrate that ferroptosis takes part in the process of UC.

L366: ..performed low scores ..

Response: As expected, DSS+Fer-1 mice performed a lower low scores of DAI and CMDI than in DSS group.

L367: … biomarker ACSL4 was decreased..

Response: The symbol expression of ferroptosis biomarker ACSL4 was declined decreased and GPX4 was increased by Fer-1 in DSS+Fer-1 group, compared with DSS group.

L392: ..indicating that high level..

Response: There was no significant difference between ACSL4+/+LPS and ACSL4+/+LPS+VD groups, indicating the that high level of ACSL4 may inhibits the relieving effect of VD on ferroptosis in some degree.

L401: that VD may alleviate ..

Response: Therefore, we speculated that the role of VD to may alleviate the pathological damage of UC partly by downregulating ACSL4 gene.

Thank you again for your suggestion and we will enhance the accuracy of the article's expression.

Sincerely yours,

Juan Kong
